# Abscopal Effect, Extracellular Vesicles and Their Immunotherapeutic Potential in Cancer Treatment

**DOI:** 10.3390/molecules28093816

**Published:** 2023-04-29

**Authors:** Aleli Salazar, Víctor Chavarria, Itamar Flores, Samanta Ruiz, Verónica Pérez de la Cruz, Francisco Javier Sánchez-García, Benjamin Pineda

**Affiliations:** 1Neuroimmunology and Neuro-Oncology Unit, National Institute of Neurology and Neurosurgery “Manuel Velasco Suárez”, Mexico City 14269, Mexico; aleli.salazar@innn.edu.mx (A.S.); vchavarria@innn.edu.mx (V.C.);; 2Immunoregulation Lab, Department of Immunology, Instituto Politécnico Nacional, Mexico City 11340, Mexico; 3Neurobiochemistry and Behavior Laboratory, National Institute of Neurology and Neurosurgery “Manuel Velasco Suárez”, Mexico City 14269, Mexico

**Keywords:** cancer, extracellular vesicles, abscopal effect, DAMPs

## Abstract

The communication between tumor cells and the microenvironment plays a fundamental role in the development, growth and further immune escape of the tumor. This communication is partially regulated by extracellular vesicles which can direct the behavior of surrounding cells. In recent years, it has been proposed that this feature could be applied as a potential treatment against cancer, since several studies have shown that tumors treated with radiotherapy can elicit a strong enough immune response to eliminate distant metastasis; this phenomenon is called the abscopal effect. The mechanism behind this effect may include the release of extracellular vesicles loaded with damage-associated molecular patterns and tumor-derived antigens which activates an antigen-specific immune response. This review will focus on the recent discoveries in cancer cell communications via extracellular vesicles and their implication in tumor development, as well as their potential use as an immunotherapeutic treatment against cancer.

## 1. Introduction

Cancer is defined as the abnormal growth of aberrant cells that can arise from any organ or body structure after evading endogenous control mechanisms [1]. It is considered as one of the most dreaded and concerning diseases of the 21st century, because the lifetime risk of cancer is estimated to be around 40% in the general population [2]. Nowadays, cancer is one of the leading causes of death worldwide, with more than 8.2 million cancer-related deaths and the highest death rates in North America, Oceania and Europe. It is estimated that about 14.1 million new cancer cases will be diagnosed worldwide in this year alone, with prostate, breast, lung, colorectal, liver and stomach being the most frequently diagnosed cancer cases [3]. Cancer causes are multifactorial; many lifestyle-associated risk factors such as tobacco use, physical inactivity, excess body weight and a sedentary lifestyle, as well as exposure to ionizing radiation and chemicals and circadian cycle disruptions, have been associated with the current prevalence [4,5]. Nowadays, the 5-year overall survival rate for cancer patients is around 30% [6].

Cancer treatment depends on the type of cancer, progression status, age and sex of the patient, among others, but generally, it consists of surgical resection, chemotherapy and radiotherapy. Recently, the use of cancer-specific medication, such as hormonal therapy or immunotherapy, with the anti-tumor vaccines and checkpoint immune blockade being the most used therapies, has shown an increase in the survival rate of cancer patients [7,8]. Despite these efforts, the current treatments have not been completely successful in the eradication of several tumors. One of the main reasons for the aggressiveness of cancer is the ability of tumor cells to communicate with each other and with the surrounding environment. It has been demonstrated that the communication between cancer, stromal and immune cells plays a fundamental role in the maintenance and progression of a tumor: a cancer cell is able to send signals to other cells via extracellular vesicles (EVs) by activating the pathways that accelerate tumor growth, generating resistance to the treatment and promoting immune evasion [9,10]. Therefore, a better understanding of the communication process between cancer cells and its implications in tumor development can lead to a potential new therapeutic target to generate successful novel treatments against cancer. In this line, it has been proved that irradiated tumor cells increase the release of EVs loaded with antigens that can activate an anti-tumor immune response. The release of EVs by irradiated cancer cells could stimulate a local and systemic immune response dependent on the tumor-derived antigens and damage-associated molecular patterns (DAMPs) contained by the EVs, promoting immunogenic cell death via the recognition of specific antigens in metastatic cancer cells [11,12]. This review will focus on the recent discoveries in cancer cell communications via EVs and their implication in tumor development, as well as their potential use as a treatment against cancer.

## 2. The Abscopal Effect in Cancer

With the use of radiotherapy in cancer, scientists have observed that the effect of whole-body irradiation was species- and dose-dependent, showing that the higher the dose, the greater the damage and the lower the repairing capacity. Later, in vitro studies revealed that radiation impaired cell proliferation due to cell-cycle arrest and chromosome damage. In vivo studies showed that bone marrow cells with a high mitotic rate died shortly after whole-body irradiation; however, while cells with a low mitotic rate were affected by irradiation in the long term, this late effect was unrelated to cell proliferation [13,14].

The concept of the abscopal effect (AE) emerged with the idea of the indirect effect of radiation, in which the radiation does not only affect one kind of cell, but it has repercussions for the whole body. This means that specific-site irradiation would produce an effect at a distant site away from the irradiated tumor tissue, inducing the regression of metastatic tumors [15]. This phenomenon was named the abscopal effect (‘ab-’ is a prefix with the meaning “position away from” and ‘scopos’ (Latin) is a mark or target for shooting, described by RH Moles in 1953. He reported on patients with tumor regression in non-treated secondary tumors after local irradiation of the primary tumor [16]).

The AE has attracted the attention of researchers for the possibility of reducing metastasis by enhancing the AE with immunotherapy. In this new century, several in vivo assays have been conducted, showing that the AE induced by RT either alone or in combination with immunotherapy elicits an effective anti-tumoral response with metastasis reduction [17,18]. AE induction depends on the dose of radiation and the eliciting of immunogenic cell death [19,20], leading to the release of DAMPs and tumor-derived antigens [21] (Table 1).

In addition, the AE is characterized by the release of EVs by cancer cells. The term EV is applied to all of the subtypes of cell-secreted membranous structures, including apoptotic bodies (ABs), ectosomes (Ecs), exosomes (Exs), dense nuclear vesicles, microvesicles (MVs) and large oncosomes. These EVs are released in response to the stress caused by irradiation that generates DNA damage, and irradiated cells enhance the release of different types of EVs into the extracellular microenvironment in a p53-dependent manner. After DNA damage, p53 is activated, thus acting as a transcription factor, inducing the direct transcription of Tumor Suppressor-Activated Pathway 6 (TSAP6), the protein associated with the release of EVs [30,31].

In the AE, the radiation of a primary tumor causes the release of soluble DAMPs [22] or the release of EVs loaded with DAMPs [12], leading to the recruitment, activation and proliferation of cells with anti-tumor activity such as dendritic cells (DCs), natural killer cells (NK cells), cytotoxic CD8+ T cells (Tc cells) and helper CD4+ T cells (Th cells), and to the inhibition of regulatory T cells (Treg cells), M2-like tumor-associated macrophages (M2-TAMs), myeloid-derived suppressor cells (MDSCs) and other cells with suppressive activity, which have a direct effect on secondary tumors (Table 2).

## 3. Extracellular Vesicles: Main Characteristics and Their Role in Cancer Progression

EV formation is dependent on the site of cellular origin; EVs are secreted by most cells from several tissues and can be found in a variety of body fluids including cerebrospinal fluid (CSF), urine, saliva, synovial fluid, bile, plasma, amniotic fluid, breast milk, semen and ascites fluid [38]. EVs play a fundamental role in intercellular communication [39,40,41,42] by delivering information via the extracellular space, thus regulating biological processes such as gene transcription, metabolism, cell migration and polarization and mediating the immune response [43,44,45].

The formation of exosomes depends on the formation of multivesicular bodies, which are early endosomes that present the inward budding of the membrane into the lumen, carrying cytoplasmic content and expressing several molecules on their surface. These multivesicular bodies are transported to the plasmatic membrane regulated by the GTPase Ras-associated Binding Protein (RAB) and the Soluble N-ethylmaleimide-sensitive Factor Adaptor Protein Receptor (SNARE) complex, and released after fusion with the plasmatic membrane via an endosomal sorting complex required for transport (ESCRT)-dependent or ESCRT-independent mechanism [45]. Meanwhile, MVs are reported to be formed via actomyosin-based membrane budding involving ADP-Ribosylation Factor 6 (ARF6), released directly from the plasmatic membrane [46].

EVs can be divided according to their size [47]: EVs ranging from 50 to 150 nm are classified as small exosomes, MVs range from 100 to 1000 nm, ABs range from 500 to 4000 nm and large oncosomes range from 1000 to 10,000 nm. The content of EVs is diverse and depends on cell type, physiologic conditions and cell damage [48], but, generally, all EVs contain several proteins, DNA, messenger RNA (mRNA), micro-RNA (miRNA), long non-coding RNA (lncRNA), lipids or lipid rafts and whole organelles or parts of them [49,50,51,52]. The EV cargo can establish intercellular communication, stimulating the response of neighbor cells. Reports suggest that EVs have therapeutic effects in heart repair, liver disease, kidney disease, wound healing, autoimmune disorders and neurodegeneration, spinal cord injury, diabetes and, particularly, cancer [53].

EV release has also been observed in the tumor microenvironment (TME); reports indicate that cancer cells release more EVs than non-malignant cells, and, interestingly, their protein content and size are different, possibly enhancing intracellular communication [54], favoring tumor progression and even leading to chemotherapy resistance and metastasis [55]. The increase in EV release in cancer cells can be induced by both intrinsic and extrinsic signaling. For instance, the activation of oncogenic signaling pathways such as the Epidermal Growth Factor Receptor variant III (EGFRvIII) pathway and the overexpression of both Pyruvate Kinase M2 (PKM2) and Harvey rat sarcoma virus (HRAS) increase EV production in cancer cells [56]. In addition to mutations and self-regulated mechanisms, the microenvironmental conditions such as hypoxia may up-regulate EV release via the Hypoxia Inducible Factor 1 α (HIF-1α)-dependent expression of RAB22 and the phosphorylation of the Proline-Rich AKT Substrate of 40 kDa (PRAS40) [41]. Additionally, oxidative stress promotes EV release in cancer and mesenchymal stem cells (MSC) [57]. Once the EVs are in the extracellular milieu, they can interact with molecules expressed on the cell surface of neighboring healthy cells, or they can be internalized via endocytosis, phagocytosis or membrane fusion [58], releasing the EV content [59] and potentially inducing structural and functional modifications of the target cell.

## 4. Interaction of Tumor-Derived Extracellular Vesicles with Non-Cancer Cells

The interaction of tumor-derived EVs with the healthy neighboring cells promotes the establishment of the tumor, further growth and expansion. This interaction has been particularly documented between tumor cells and fibroblasts, as well as endothelial, mesenchymal and immune cells [60] (Figure 1).

Fibroblasts are one of the most abundant types of non-cancer cells in the TME, participating in the deposition and remodeling of the extracellular matrix. After pathological activation by cancer cells, fibroblasts become part of the tumor, changing their phenotype to cancer-associated fibroblasts (CAFs). In this regard, tumor-derived EVs can induce the pathological transformation of CAFs, particularly in the early stages of the carcinogenesis [61]. The activation of CAF-related genes is induced by the signature cargo in tumor-derived EVs, such as Transforming Growth Factor beta (TGF-β), IL-1, IL-6 and miRNAs (miR-155-5p, miR-1247-3p and miR-211), activating the Smad2/3 and Smad4 pathway, the JAK–STAT pathway, the mitogen-activated protein kinase (MAPK) pathway and the activation of the nuclear factor kappa-light-chain-enhancer of activated B cell (NFκB) transcription factor [62,63], leading to the CAF synthesis of growth factors, cytokines and soluble factors that will promote fibrosis, angiogenesis, escape from immune surveillance, tumor growth and metastasis [64]. On the other hand, tumor-derived EVs can also interact with endothelial cells which have a pivotal role in tumor development in terms of the generation of new blood vessels. The generation of blood vessels from pre-existing vascular networks helps the maintenance of tumor growth by supplying the nutritional and oxygen requirements of the tumor. Several studies have proven that tumor-derived EVs can promote angiogenesis by delivering growth factors such as vascular endothelial growth factor (VEGF), platelet-derived growth factor (PDGF) and angiopoietin 1 [65]; miRNAs such as miR-9 [66], miR-23a [67], miR-135b [68,69], miR-494 [70], miR-1246 [71] and miR-210 [72]; lncRNAs such as H19 [73], MALAT1 [74], CCAT2 [75] and POU3F3 [76] and proteins such as Wnt4, which activates the β-catenin pathway in endothelial cells [77,78].

## 5. Interaction of Tumor-Derived Extracellular Vesicles with Immune System

Tumor-derived EVs can regulate TME and serve as suppressors in the immune response against tumors [79,80], via the regulation of phagocytosis, antigen presentation, cytokine synthesis and cytotoxicity against tumor cells, impairing the function of macrophages and their precursors such as monocytes, T cells, NK, DCs, cells and B cells [81,82,83], as described below.

### 5.1. Monocytes

Macrophages are cells commonly found in TME and have been associated with tumor progression. Depending on the gene expression, cytokine synthesis and effector activity macrophages can be polarized to the M1 macrophage with anti-tumoral activity, or the M2 macrophage with protumoral activity. Inside the TME, macrophages can be recruited and polarized to an M2-like phenotype via the action of tumor-derived EVs, and they are known as tumor-associated macrophages (TAMs) which promote tumor growth and metastasis [84,85]. M2-TAM polarization can also be induced by lncRNA contained in tumor-derived EVs, such as HCG18 [86] and HPPR1 [87]. Likewise, lnc-TALC has been related to chemotherapy resistance due to its capacity to promote the expression of the DNA repair enzyme O6-methylguanine-DNA methyltransferase (MGMT) via the modulation of histone H3 acetylation. Additionally, tumor-derived EVs containing lnc-TALC can promote the M2 polarization of microglia by increasing the expression of Arginine-1 (Arg-1) and CD163, as well as M2-related cytokines (TGF-β, IL-4 and IL-10), via p38 of the MAPK pathway activation [88]. Several miRNAs carried by tumor-derived EVs can induce the M2 polarization, such as miR124, by inhibiting the TLR4 signaling transduction [89]; likewise, miR-934 [90] and miR-27a-3p upregulate Programed Cell Death Ligand 1 (PD-L1) expression by inhibiting the Phosphatase and Tensin homolog (PTEN) activity on the Phosphatidylinositol 3-Kinase (PI3K)-Akt signaling pathway. The PD-L1 expression on M2-TAMs inhibits Tc cell effector functions and promotes tumor immune evasion [91]. Another tumor-derived EV mechanism associated with M2 polarization is the expression of the IL-6 receptor subunit β (gp130) on their membrane surface which can be transferred to bone-marrow-derived macrophages (BMDMs). The signaling by gp130 induces the phosphorylation of Signal Transducer and Activator of Transcription 3 (STAT3) and promotes the M2 phenotype, leading to changes in the expression of cytokines, such as IL-6, IL-10 and CXCR4, and the overexpression of CCL2 mRNA [92]. Tumor-derived EVs also affect the macrophage precursors. For instance, there are reports indicating that cancer-stem-cell (CSC)-derived EVs tend to accumulate inside the CD14+ monocytes more efficiently than in other immune cells, inducing cytoskeletal restructuring, generating filopodia and triggering diapedesis. Additionally, CSC-derived EVs induce the M2 polarization of monocytes, increasing the expression of the immunosuppressive molecule PD-L1, M2 markers CD163 and CD206 and monocyte-recruiting chemokines Monocyte-Chemotactic Protein 3 (MCP-3 or CCL7) and C-X-C motif Chemokine Ligand 1 (CXCL1), as well as IL-1β, IL-6, IL-10 and arginase-1, leading to the suppression of T cell proliferation [93,94].

### 5.2. Dendritic Cells

Tumor-derived EVs promote the expansion of myeloid-derived suppressor cells (MDSCs), an immature myeloid cell type with the ability to suppress T cell activation. The promotion of MDSCs via an IL-6-dependent mechanism can be sustained by the binding of Heat Shock Protein 72 (HSP72), expressed at the surface of tumor-derived EVs, to Toll-Like Receptor 2 on myeloid cells, and the posterior activation of the Myeloid Differentiation primary response 88 (MyD88) pathway, consequently inducing IL-6 secretion and maintaining the MDSC phenotype in an autocrine manner [95].

Previous reports have indicated that small tumor-derived EVs, containing galectin 9 (LGALS9), can impair DC antigen presentation via downregulating the expression of HLA-A, CD40, CD86 and Transporter Associated with Antigen Processing 1 (TAP1) proteins, contributing significantly to tumor progression and poor prognosis [96]. Tumor-derived EVs that contain miR-203 can inhibit DC maturation via the downregulation of the expression of TLR-4 and Tumor Necrosis Factor alpha (TNF-α), impairing the capacity to mediate IL-12-dependent Th1 differentiation [97]. Tumor-derived EVs containing miR-424 inhibit the expression levels of costimulatory molecule CD80 expressed on DCs and the expression of CD28 on Th cells and Tc cells, impairing T cell activation and proliferation [98].

### 5.3. T Cells

Moreover, tumor-derived EVs impair lymphocyte function, since they carry different immunosuppressive molecules such as death receptor Fas, Fas ligand (FasL), TNF-α, TNF-Related Apoptosis-Inducing Ligand (TRAIL), Cytotoxic T Lymphocyte Antigen 4 (CTLA-4), Programed Cell Death Protein 1 (PD-1), PD-L1, CD39 and CD73. It has been reported that in several types of cancer the expression of these molecules is directly correlated with the later stages of the disease [99,100]. Tumor-derived EVs carrying cell death receptors and their ligands can induce apoptosis on T cells by activating caspases 3 and 7 [101,102]; additionally, tumor-derived EVs can decrease the synthesis of TNF-α and Interferon gamma (IFN-γ) via Tc cells in a dose-dependent manner [103]. Particularly, tumor-derived EVs expressing PD-L1 can bind to PD-1 expressed on the surface of T cells, impairing T cell activation via anti-CD3 antibody ligation and DC antigen presentation [100]. The regulation of T cells can be mediated by the induction of the CD39-CD73 axis and ectonucleotidases involved in the dephosphorylation of ATP into adenosine, inducing a CD4+ FoxP3+ Treg phenotype and decreasing the synthesis of IL-2 and TNF-α [104]. Therefore, tumor-derived EVs containing CD73 can inhibit granzyme B and INF-γ synthesis in Tc cells [105]. Moreover, CSC-derived EVs can reduce T cell activation and proliferation via decreasing the expression of the alpha chain of the IL-2 receptor (CD25) and the late activation marker CD69. Therefore, EVs can impair the synthesis of Th1-proinflammatory cytokines, such as IL-2, INF-γ and TNF-α [93].

### 5.4. NK Cells

NK cells can be altered by tumor-derived EVs containing immune checkpoint molecules PD-1, PD-L1 and CTLA-4, which impair their cytotoxic activity by downregulating the expression of Natural Killer Group 2 Member D Protein (NKG2D) on the surface of NK cells [99,103]. NKG2D is an activating receptor expressed on NK cells, triggering cytotoxic activity and cytokine secretion after the ligation of MHC-class I chain-related A and B (MICA/MICB) and UL-16 Binding Protein (ULBP) [106]. Tumor-derived EVs also express the surface ligands of NKG2D, such as ULBP-2 and MICA/MICB, acting as negative regulators of NKG2D on NK cells in a dose-dependent manner and impairing cytotoxicity against tumor cells [107].

### 5.5. B Cells

B cells can be polarized to a regulatory phenotype via the tumor-derived EV cargo. In this context, HMGB1 expressed on the surface of tumor-derived EVs can interact with TLR-2 on B cells, inducing the expression of T cell Immunoglobulin and Mucin domain 1 (TIM-1) on B cells via an MAPK-pathway-dependent mechanism. The presence of TIM-1+ B cells decreases the expression of TNF-α and the IFN-γ production of Tc cells and correlates with tumor progression and short survival time [108,109]. Another mechanism to inhibit B cell function via tumor-derived EVs is related to the inhibition of phosphorylated Bruton’s Tyrosine Kinase (p-BTK), reducing the proliferation and viability of B cells [110]. The BTK protein mediates actin remodeling via the regulation of the Wiskott–Aldrich syndrome protein (WASP), the protein involved in downstream BCR signaling; the reduction in p-BTK impairs BCR-mediated antigen internalization and consequently impairs BCR-mediated antigen presentation [111].

## 6. Extracellular Vesicles Mediating the Abscopal Effect

As described previously, tumor-derived EVs can affect the cells surrounding the tumor promoting their proliferation via immune system evasion. However, various research groups are taking advantage of tumor-derived EVs and non-cancer-cell-derived EVs to design therapeutic strategies against the tumor. EVs carry different molecules with immunostimulatory and immunosuppressive functions, driving the immune response. Radiation induces the release of EVs enriched with molecules that activate an anti-tumor immune response, by modulating the antigen presentation, activation and proinflammatory polarization of immune cells; thus, EVs have gained the spotlight as a potential form of immunotherapy against tumors [11,12,112,113].

EVs are small organelles released to extracellular space as a mechanism of cellular communication. When tumor cells are irradiated, a communication mechanism is initiated between the irradiated tumor cells and the surrounding cells via the release of EVs, which promote bystander and abscopal effects [114]. The bystander effects are promoted via the amount of bioactive molecules contained in the EVs, which upon entering neighboring cells induce epigenetic changes and cause DNA damage resulting in decreased proliferation and apoptosis [115,116,117,118]. The AE is mainly attributed to the potentiation of the anti-tumor immune response induced by the antigens contained in the EVs, a characteristic that has been exploited to use EVs derived from irradiated cells in an anti-tumor vaccine [12,54].

## 7. Modifying Extracellular Vesicles for Therapeutic Purposes

Beyond mediating tumor progression, EVs could be potentially used as therapeutic tools. In this line, several studies [119] have shown that EVs have high bioavailability in tumor tissue and can act as carriers to deliver different molecules such as DNA, RNA, microRNAs, proteins and lipids [80], which make EVs capable of modifying the TME via several mechanisms that enhance anti-tumor activity or could potentiate the current therapeutic approach [120,121].

### 7.1. Anti-Tumoral Effects of Modified Extracellular Vesicles

EVs can suppress tumor progression via modifications to the molecules expressed on their surface and inside them, independently of their origin. In line with this approach, colorectal cancer cells treated with starved tumor-cell-derived EVs loaded with miR-34a showed the inhibition of both proliferation and migration, accompanied by apoptosis, via the downregulation of IL-6R, STAT3, PD-L1 and VEGF-A expression in vitro, with prolonged survival time and impaired immune evasion in a solid tumor [122,123]. Likewise, when starved tumor-cell-derived EVs are loaded with miR-124, they induce apoptosis of colorectal cancer cells and modify the immunosuppressive microenvironment of the tumor, achieving the suppression of tumor growth and an increase in survival time [124]. In a similar way, the inhibition of the epithelial-mesenchymal transition in endometrial cancer cells has been achieved via treatment with miR-192-5p expressed in TAM-derived EVs, via the inhibition of the Interleukin 1 Receptor Associated Kinase 1 (IRAK1)/NFκB signaling pathway [125]. On the other hand, the silencing of Sirtuin 6 (SIRT6) in prostate cancer cells via SIRT6-siRNA expressed in HEK 293 cell-derived EVs decreases tumor cell proliferation and metastasis via a NOTCH-related mechanism [126]. Similarly, EVs derived from adipose-tissue-derived MSC (AT-MSC) loaded with the long non-coding RNA maternally expressed gene 3 (lncRNA MEG3) inhibit tumor cell proliferation and tumor growth via a mechanism dependent on PTEN activation and the inhibition of the β-catenin pathway [127,128].

Another mechanism associated with the anti-tumor activity of EVs is the sensitization to chemotherapy; for example, AT-MSC-derived EVs have been modified to express miR-122, showing that the treatment of hepatocellular carcinoma cells with these EVs increases the sensitivity of tumor cells to sorafenib [129], since miR-122 inhibits the expression of the multidrug-resistance-related genes, such as ATP-binding cassette (ABC) transporters [130]. Similarly, miR-199a-3p-expressing AT-MSC-derived EVs induce sensitivity to doxorubicin via the downregulation of mTOR and p21-activated kinase 4 (PAK4) in hepatocellular carcinoma cells [131,132]. Likewise, treatment with EVs isolated from Carnitine palmitoyltransferase I (CPT1A) siRNA-transfected HEK 293 cells allows for the reversal of oxaliplatin resistance in colon cancer cells via the inhibition of fatty acid oxidation [133]. In this regard, the inhibition of protumoral miR-21 has been linked to the sensitization of tumor cells to several drugs. So, a treatment with blood-derived EVs with a coating of an inhibitor of miR-21 decreased tumor growth and induced sensitivity to doxorubicin in an in vivo glioblastoma model dependent on the inhibition of Akt phosphorylation [134]. Additionally, the treatment with EVs derived from HEK 293 cells loaded with a miR-21 inhibitor and coated with a Human Epidermal Growth Factor Receptor 2 (HER2) affibody decreased tumor growth via the activation of PTEN and the decreased resistance to 5-fluorouracil in colon cancer cells [135].

### 7.2. DC Activation via Extracellular Vesicles

Tumor-derived EVs are a large source of tumor-associated antigens (TAAs) and neoantigens (NAs) that could be used as therapeutic vaccines to potentiate a specific anti-tumor memory response dependent on DCs, as seen in Figure 2. Therefore, the recognition of tumor-derived EVs has been enhanced by the addition of high-glycans as sialic acid on their surface, promoting their capture by DCs via the Dendritic-Cell-Specific Intercellular adhesion molecule-3-Grabbing Non-integrin (DC-SIGN) receptor, inducing the increased processing of tumor antigens and their expression on MHC class I and II molecules [136,137]. Likewise, the expression of CD40L on tumor-cell-derived EVs promotes the binding to CD40 expressed on DCs, inducing their uptake and enhancing the tumor antigen presentation of T cells [138]. Another mechanism to promote DC activation via tumor-derived EVs is by loading EVs with miRNAs implicated in the DC response, such as let-7i, miR-155 and miR-142, inducing DCs with a mature phenotype and enhancing the expression of co-stimulatory molecules and tumor antigen presentation [139] associated with T cell activation and a decrease in tumor volume [140].

Historically, the ability of microorganisms to activate a proinflammatory immune response by binding to pattern recognition receptors (PRRs) has been used as an anti-tumor tool. In concordance with the activation of DCs, several studies have focused on pathogen-associated molecular patterns (PAMPs) to enhance the proinflammatory immune response and promote anti-tumoral activity via the association with tumor-derived DAMPs [141]. In this regard, tumor-derived EVs have been loaded with the Early Secretory Antigenic Target (ESAT-6), derived from *Mycobacterium tuberculosis*, which has been shown to be a successful inducer of DC maturation and is associated with a proliferative response of lymph node lymphocytes and a decrease in tumor volume [142,143]. Likewise, tumor-derived EVs containing microbial unmethylated cytosine-phosphate-guanosine (CpG) DNA are highly effective at activating DCs and promoting the processing and presentation of tumor antigens, as well as increasing the specific humoral IgG antibody response [144].

The activation of DCs with tumor-derived EVs has been classically directed against tumor antigens; however, the immune response can be targeted against other cells in the TME. This is the case for CAFs, which have been targeted via the expression of fibroblast activation protein α (FAP-α), where tumor-derived EVs loaded with tumor antigens and expressing FAP-α induce the maturation of DCs, increasing the infiltration of specific T cells against tumor cells and FAP-positive fibroblasts, and reducing the proportion of M2-TAMs, MDSCs and Treg cells in the in vivo tumor models of murine colon carcinoma, melanoma, Lewis lung carcinoma and breast cancer, with a mechanism associated with the induction of tumor cell ferroptosis and IFN-γ synthesis [145].

### 7.3. TAM Reprograming via Extracellular Vesicles

Given the important role of TAMs in the progression of cancer, EVs have been modified with the intent of reprograming the M2-TAM polarization toward an M1 phenotype with anti-tumor activity. In this line, macrophage-derived EVs loaded with metformin and coated with mannose to give affinity for M2-type macrophages via the CD206 receptor allowed for the reprograming of M2-TAM to the M1-like phenotype via an AMPK-NF-κB-dependent mechanism, remodeling the immune microenvironment by increasing Tc cell infiltration and reducing the amount of MDSCs and Treg cells [146,147].

Another approach aimed at modifying the phenotype of M2-TAMs is the use of bone-marrow-derived MSC (BM-MSC)-derived EVs, loaded with siRNA-galectin 9 and oxaliplatin. Galectin 9 is expressed by tumor cells and is a ligand of dectin 1, whereby the galectin 9/dectin 1 axis induces the polarization of macrophages toward an M2 phenotype [148]; therefore, the silencing of galectin 9 suppresses the generation of protumoral macrophages, and the combination with oxaliplatin induces immunogenic death, enhancing the immune response. The combined action of these mechanisms by means of BM-MSC-derived EVs modifies the TME in a pancreatic cancer model, increasing the proportion of M1-type macrophages, infiltrating Tc cells and decreasing the amount of M2-TAMs and Treg cells, reflected in the decrease in the tumor volume and the increase in the survival time [149].

The reprograming of M2-TAMs to the M1 phenotype can also be achieved via the inhibition of the main transcription factors associated with the M2 phenotype. In this regard, the treatment with HEK 293 cell-derived EVs loaded with antisense oligonucleotides targeting STAT6 is highly effective at modifying the TME reprograming of M2-TAMs to the M1 phenotype in a model of colorectal carcinoma and a model of hepatoma, with increased expression of inducible nitric oxide synthase (iNOS), IL-1β, IL-12 and TNF-α [150].

The modification of the tumor extracellular matrix is another way to reprogram M2-TAMs. In this sense, HEK 293 cell-derived EVs expressing hyaluronidase and coated with folic acid show a high accumulation in the tumor bed, where high-molecular-weight hyaluronic acid is degraded to low-molecular-weight hyaluronic acid, which is associated with the polarization of macrophages to an M1 phenotype, with the consequent modification of the TME, lowering the percentage of M2-TAMs and Treg cells and the expression of IL-4 and IL-10 and increasing the expression of TNF-α and IL-6 and the infiltration by Tc cells [151].

The proinflammatory effect of DAMPs can also induce the reprograming of M2-TAMs to the M1 phenotype. In this regard, irradiated tumor-cell-derived EVs containing large amounts of DAMPs increased the expression of iNOS, IL-1, IL-6, IL-12, TNF-α and IFN-γ and modified the TME. Additionally, EVs derived from irradiated tumor cells also induce PD-L1 expression in TAMs; however, their anti-tumor effect can be enhanced via the addition of anti-PD1 antibody, inducing the generation of memory T cells [152].

The interplay of PAMPs and the proinflammatory response can enhance the reprograming of M2-TAMs. In this context, reports have described that EVs derived from *Toxoplasma gondii*-infected DCs contain a large amount of miR-155-5p, and can modify the TME by decreasing the proportion of MDSCs within the tumor and inducing the polarization of peripheral macrophages to M1 phenotype, with increased expression of iNOS and TNF-α that are directly associated with decreased tumor volume and increased survival time. This effect is due to the potential targets of miR-155-5p, such as Suppressor Of Cytokine Signaling 1 (SOCS1), since inhibiting SOCS1 prevents its suppressive effect on the polarization of macrophages to the M1 phenotype [153,154].

The inability to phagocytose tumor cells is another protumoral mechanism associated with M2-TAMs. In this context, the CD47-SIRPα axis is a “don’t eat me” signal involved in the inhibition of the phagocytic activity of macrophages. So, phagocytosis can be enhanced by the blockade of the CD47-Signal Regulatory Protein α (SIRPα) axis using different molecules to prevent the interaction between CD47 expressed on tumor cells and SIRPα expressed on macrophages. In this context, EVs obtained from HEK 293 cells transfected to express SIRPα can accumulate in the tumor niche, inhibiting tumor growth by enhancing the phagocytosis of CD47+ tumor cells and increasing the infiltrating Tc cells [155]. Similarly, EVs obtained from M1-type macrophages coupled to the Fc region of neutralizing antibodies against CD47 and antibodies against SIRPα blocked the signaling of the CD47-SIRPα axis, reprogramed M2-TAMs to the M1 phenotype and induced the expression of CD80, CD86 and iNOS [156] (Figure 3).

### 7.4. Stimulation of T Cell Response via Extracellular Vesicles and Their Interplay with Immune Checkpoint Inhibition

T cells play a pivotal role in the specific anti-tumor immune response, acting as effectors of the immune memory against tumor-derived antigens. To elicit a T cell response, DCs must process and present antigens to properly activate T cells, and this depends on the adequate signaling of the CD28-CD80/86 co-stimulation axis. In this line, treatment with miR-424 KO tumor-derived EVs induces the expansion of T cells against tumor antigens and increases the effectiveness of anti-PD-1/CTLA-4 antibodies, evidenced by the inhibition of tumor growth and metastasis, and the increase in median survival in a colorectal cancer model, since miR-424 is a negative regulator of the CD28-CD80/86 axis [157].

Some researchers have focused their interest on promoting the direct interaction between T cells and tumor cells to facilitate the generation of a T cell response. In this regard, the generation of EVs derived from HEK 293 cells transfected to express anti-CD3-anti-HER2 bispecific single-chain variable fragment (scFv) antibodies on their surface has facilitated the interaction between HER2-expressing breast cancer cells with CD3-expressing T cells; this interaction efficiently activates T cells, induces the apoptosis of tumor cells and causes the inhibition of tumor growth in an in vivo model associated with the increased infiltration of T cells [158].

The anti-tumor efficiency of chimeric antigen receptor (CAR) T cells can also be improved by inducing the expression of the RNA Component Of Signal Recognition Particle 7SL1 (RN7SL1), secreted via EVs from CAR-T cells. RN7SL1 loaded in CAR T-cell-derived EVs promotes the maturation of peripheral DCs, increasing the expression of MHC class II and co-stimulatory molecules, and also restricting the development of MDSCs and downregulating the expression of TGF-β, enhancing tumor cell death via CAR-T cells and tumor-specific endogenous T cells [159]. Another advantage of CAR-T-cell-derived EVs is their intrinsic and specific cytotoxic activity on tumor cells. This approach has been evidenced using mesothelin-specific CAR-T-cell-derived EVs against breast cancer cells, mediating the killing of tumor cells in vitro and in vivo by the expression of perforin and granzyme B [160]. Similarly, EVs derived from EGFR-specific and HER2-specific CAR-T cells are cytotoxic against breast cancer cells in vitro and in vivo, showing the advantage of lacking the expression of PD-1, so that they cannot be inhibited by the PD1-PD-L1 axis [161].

Following the importance of the immune checkpoint PD-1-PD-L1 axis on the T cell response, some researchers have focused on blocking this axis to improve the fitness of the tumor-infiltrating T cells. Characteristically, activated T cells overexpress PD-1 on their surface; so, the use of EVs derived from activated T cells expressing PD-1 is effective in blocking the dysfunction generated by the ligation of PD-L1 expressed by tumor cells, promoting the synthesis of IL-2, IFN-γ and granzyme B in infiltrating T cells in a breast cancer model [162]. Similarly, the silencing of PD-L1 expression in tumor cells via treatment with human neural progenitor cell-derived EVs modified with PD-L1 siRNA and with a brain-tumor-targeting cyclic RGDyK peptide increases tumor-infiltrating EVs, decreases PD-L1 expression by glioblastoma cells and increases infiltration by Tc cells, showing a synergic effect with radiotherapy [163].

To improve the bioavailability of therapeutic antibodies aimed at inhibiting immune checkpoint molecules, HEK 293 cell-derived EVs overexpressing CD64 and the high-affinity Fc receptor of IgG have been coupled with the anti-PD-L1 antibody, improving the stability and bioavailability of the antibody to the tumor, and showing an anti-tumor effect in vivo against melanoma, with the proliferation of Tc cells and inhibition of Treg cells [157]. In this regard, the bioavailability and anti-tumor activity of the anti-PD-L1 antibody are enhanced when co-administered with hyaluronidase-coupled HEK 293 cell-derived EVs, due to the degradation of hyaluronan in the TME in a breast cancer model, inducing a memory anti-tumor immune response dependent on Tc cells [164]. It is important to emphasize that even though the PD-1-PD-L1 axis is usually focused on the T cell response, it could potentially improve the anti-tumor cytotoxic response of NK cells.

The potentiation of immune responses via the interplay of DAMPs/PAMPs with PRR expressed on immune cells has been demonstrated to be effective at developing a T cell response. In this line, a study using EVs derived from irradiated tumor cells showed the potential use as a vaccine against glioblastoma cells, since tumor-derived EVs after irradiation have a high expression of TAA and neoantigens and are loaded with a greater amount of DAMPs. Their use in an in vivo glioblastoma model induced the infiltration of activated Tc and Th cells and inhibited tumor growth [54]. Since Treg cells are some of the major immunosuppressive cells in the TME, a therapeutic target is the reprograming of Treg cells to a phenotype with anti-tumor activity. In this line, EVs derived from heat-stressed tumor cells expressing HSP70 are highly endocytosed by DCs, inducing the synthesis of IL-6 and the differentiation of Treg cells to a type 17 Th cell with the synthesis of IL-17, reducing tumor growth in a prostate cancer model [165]. Likewise, the use of EVs derived from the outer membrane of Gram-negative bacteria (GNB) genetically modified to express PD-1 are effective in modifying the TME by increasing infiltration by DCs, NK cells and Tc cells, and stimulating the secretion of IFN-γ, TNF-α and IL-6, thus generating greater effectiveness than anti-PD-L1 antibody treatment at inducing an effective anti-tumor immune response [166]. Similarly, GNB-derived EVs loaded with a PD-1 plasmid and coated with a tumor-targeting peptide induce the expression of PD-1 in tumor cells, promoting the inhibition of the PD-1/PD-L1 axis via a juxtracrine stimulus, and promoting the recruitment and activation of Tc cells and NK cells as well as the secretion of INF-γ [167], as seen in Figure 4.

## 8. Conclusions

Cellular communication via extracellular vesicles has a pivotal role in the development of cancer. Thus, this intercommunication between cells could be used for therapeutic purposes against cancer, since the use and modification of EVs have shown promising results in both in vitro and in vivo assays. Therefore, EVs could be further modified to express a variety of molecules, considering their cellular origin, their content of tumor antigens and the enhancing effect of damage- and pathogen-associated molecular patterns, to regulate the tumor microenvironment and potentially be used as a therapy to treat cancer.

## Figures and Tables

**Figure 1 molecules-28-03816-f001:**
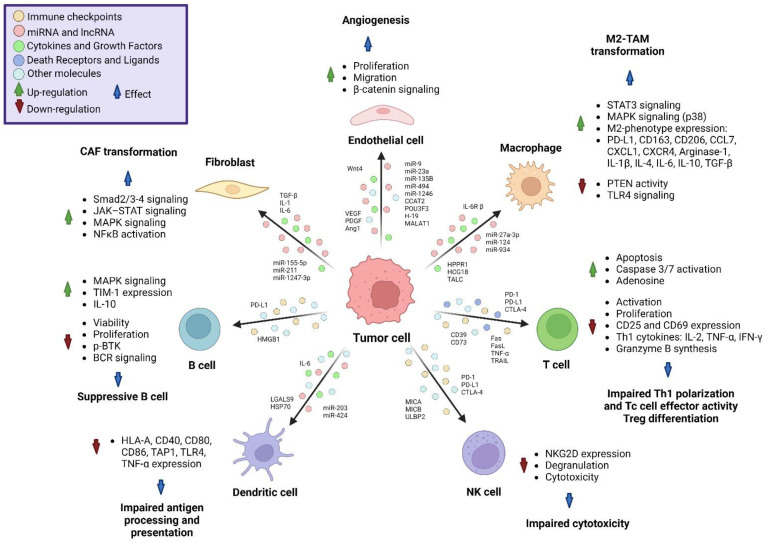
Effects of tumor-derived EV content on non-cancer cells. Tumor cells can release EVs loaded with miRNAs, LncRNA, cytokines, growth factors and immune checkpoint molecules, as well as death receptors and their ligands; the content of the EVs will differentially affect the tumor microenvironment by affecting immune, endothelial and mesenchymal cells within the tumor, promoting the activation of several pathways leading to immune escape and tumor progression.

**Figure 2 molecules-28-03816-f002:**
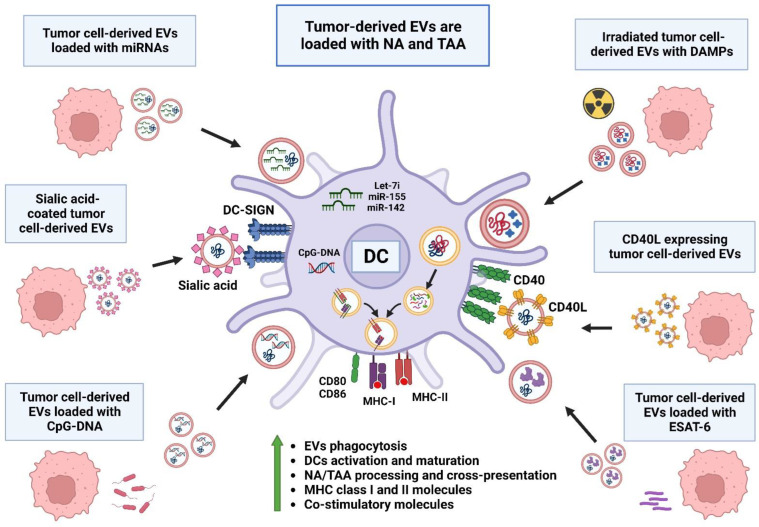
DC activation via extracellular vesicles. DC activation can be enhanced via the modification of tumor-derived EVs containing TAA and NA via several mechanisms: first, by coating the EVs with molecules such as sialic acid or CD40L to promote the uptake of EVs; second, via the loading of miRNAs related to activation and antigen processing, such as Let-7i, miR-155 and miR-142; third, via the codelivery of PAMPs in the EVs, such as ESAT-6 and CpG-DNA; and fourth, via the codelivery of DAMPs in the EVs after the irradiation of tumor cells, leading to increased EV phagocytosis, DC activation and maturation, antigen processing and cross-presentation and the expression of MHC class I and MHC class II molecules and costimulatory molecules such as CD80 and CD86.

**Figure 3 molecules-28-03816-f003:**
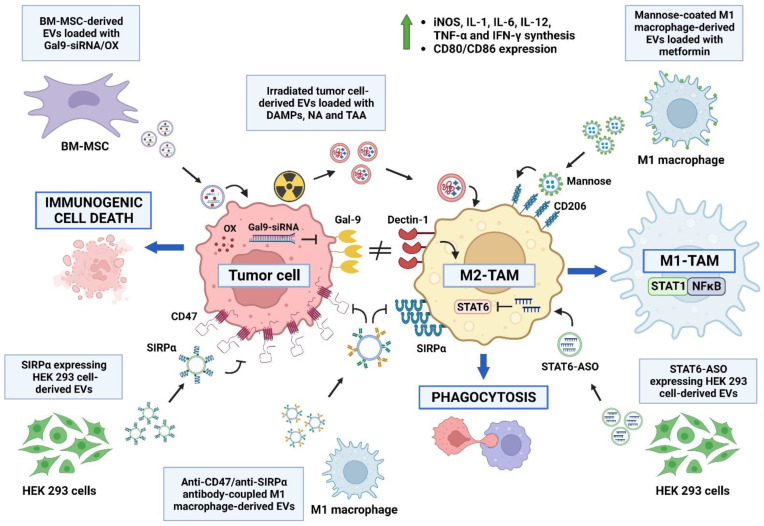
TAM reprograming via extracellular vesicles. EVs can be modified to reprogram M2-TAMs to M1-TAMs, independently of their cell origin. Irradiated tumor-cell-derived EVs loaded with damage-associated molecular patterns (DAMPs), tumor-associated antigens (TAAs) and neoantigens (NAs) induce the M1 polarization of TAMs. EVs loaded with Gal9-siRNA and oxaliplatin (OX) inhibit the expression of Gal9 in tumor cells, blocking the activation of dectin-1 on macrophages and inducing the M2 phenotype via immunogenic cell death. EVs loaded with STAT6-ASO inhibit M2 polarization by inducing STAT1. EVs coated with mannose promote their uptake via CD206-expressing M2 macrophages and induce M1 phenotype via the delivery of metformin, via the activation of NFκB. The “don’t eat me” signal by the CD47-SIRPα axis can be blocked with EVs expressing SIRPα and by EVs conjugated with anti-CD47 and anti-SIRPα antibodies, promoting phagocytosis, the synthesis of iNOS, IL-1, IL-6, IL-12, TNF-α and IFN-γ and the expression of CD80/CD86 molecules.

**Figure 4 molecules-28-03816-f004:**
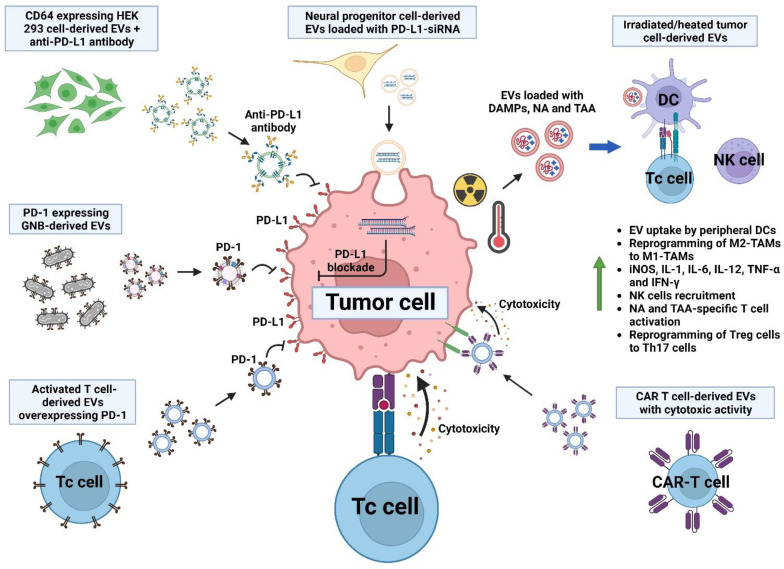
Stimulation of T cell response via extracellular vesicles and their interplay with immune checkpoint inhibition. T cell activation is enhanced via the inhibition of the PD-1-PD-L1 axis by several mechanisms. Among these are the downregulation of PD-L1 by EVs loaded with PD-L1-siRNA in tumor cells, the blockade of PD-L1 via the anti-PD-L1 antibody coupled to CD64-expressing EVs, the blockade of PD-L1 on tumor cells by Gram negative bacteria (GNB)-derived EVs expressing PD-1, and the blockade of PD-L1 via the delivery of activated T-cell-derived EVs overexpressing PD-1. Additionally, irradiated or heated tumor-cell-derived EVs effectively induce a memory T cell response. CAR-T-cell-derived EVs maintain cytotoxic activity.

**Table 1 molecules-28-03816-t001:** DAMPs released by radiation and their effect on non-cancer cells in preclinical studies.

Cancer Cell	Radiation Dose	DAMPs	Effect on Non-Cancer Cells	References
HCC1937 human breast cancer cells	Single ablative dose of 20 Gy	HSP70 HMGB1 S100A8/A9	Endothelial cell activation and surface expression of adhesion molecules ICAM-1, VCAM-1 and E-selectin and release of IL-6, CXCL1, CXCL2 and CCL7. Recruitment of neutrophils and monocytes, and differentiation and maturation of antigen-presenting cells.	[22]
Glioma stem cells	10 Gy X-ray (160 kV) at a dose rate of 0.50 Gy/min	HSP70/90 ATP HMGB1	Increased phagocytosis and DC maturation, and proliferation of T cells.	[23,24]
A549, NCI-H520 (human) and LLC lung cancer cells	100 keV, with a dose rate of 1.0 Gy/min Total dose of 6 Gy	HMGB1	Maturation and activation of DCs, and activation of memory Th cells and effector Tc cells.	[25]
MDA-MB-231 (breast), H522 (lung) and LNCaP (prostate) cancer cells	100 Gy at a dose rate of 5.56 Gy/min	CRT ATP HMGB1	Increased sensitivity to antigen-specific Tc cell lysis, and enhanced T-cell recognition of specific HLA-I.	[26,27]
HCT116 human colorectal cancer cells	5 Gy at a dose rate of 1 Gy in 63 s	HMGB1 ATP UTP	Monocyte migration.	[28]
B16F10 mouse melanoma cells	20 Gy, 120 kV	ATP HSP70 HMGB1	Upregulation of CD80, CD86, MHC-II and CD40 on DCs and proliferation of Th and Tc cells.	[29]

Abbreviations: Gy (gray), HSP (heat shock protein), HMGB1 (High-Mobility Group Box 1), S100A (S100 calcium-binding protein A), ICAM-1 (Intercellular Adhesion Molecule 1), VCAM-1 (Vascular Cell Adhesion Molecule 1), IL-6 (Interleukin 6), CXCL1 (Chemokine (C-X-C motif) ligand 1), CXCL2 (Chemokine (C-X-C motif) ligand 2), CCL7 (Chemokine (C-C motif) ligand 7), kV (kilovolts), ATP (Adenosine triphosphate), DCs (dendritic cells), keV (kiloelectronvolt), CRT (Calreticulin), Tc cells (cytotoxic CD8+ T cells), HLA-I (Human Leukocyte Antigen class-I), UTP (Uridine triphosphate), MHC-II (Major Histocompatibility Complex class-II) and Th cells (helper CD4+ T cells).

**Table 2 molecules-28-03816-t002:** DAMPs associated with the AE and their immune effect in metastasis models.

**DAMPs**	**Model**	**AE Inducer**	**Immune Effect**	**Effect over Secondary Tumor**	**References**
HMGB1	4T1 mouse breast cancer cells in BALB/c mice	24 Gy (3 × 8 Gy)	M1 polarization of macrophages, recruitment and secretion of TNF-α	Reduced tumor growth, inhibition of proliferation and migration	[32]
mtDNA ATP	B16-CD133 melanoma cells in C57BL/6N mice C51 colon carcinoma cells in BALB/c mice	24 Gy (2 × 12 Gy) + cisplatin + anti-PD-1 16 Gy (2 × 8 Gy) + cisplatin + anti-PD-1 antibody	DC recruitment and activation via secretion of ATP, mtDNA and type I IFN, Tc cell cross-priming and recruitment	Reduced tumor growth and necroptosis	[33]
CRT	LLC Lewis lung carcinoma cells in C57BL/6 mice	10 Gy (5 × 2 Gy) + Cisplatin-loaded nanoparticles + anti-PD-1	Tc cell activation, proliferation and recruitment	Reduced tumor growth and increased CXCL10 synthesis	[34]
CRT HSP-70 HSP-90	4T1 mouse breast cancer cells in BALB/cfC3H mice	Helium-driven plasma gas jet with 1 W per 300 s	Increase in tumor-infiltrating DCs and Th cells	Reduced tumor growth, apoptosis and CRT expression	[35]
CRT HMGB-1 HSP-70	TC-1 lung tumor cells in C57BL/6 J mice	24 Gy (3 × 8 Gy) + oncolytic vaccinia virus	M1 polarization of macrophage, increase in Tc and Th activation and recruitment and decrease in Treg cells	Reduced tumor growth	[36]
CRT HMGB1 HSP70	Pan02 murine pancreatic adenocarcinoma cells in C57BL/6 mice	Electroporation with 1000 V per 100 ms	Increase in effector and memory Tc cells, and Tc cell recruitment	Reduced tumor growth and decreased LOX expression	[37]

Abbreviations: TNF-α (Tumor Necrosis Factor α), mtDNA (mitochondrial DNA), PD-1 (Programed Cell Death protein 1), IFN (Interferon), CXCL10 (Chemokine (C-X-C motif) ligand 10), LOX (Lysyloxidase).

## Data Availability

Not applicable.

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
