# Peer review of "Abscopal Effect, Extracellular Vesicles and Their Immunotherapeutic Potential in Cancer Treatment"

_molecules, 2023, doi:10.3390/molecules28093816_

Round 1

Reviewer 1 Report

Review

molecules-2357010

“Abscopal effect, extracellular vesicles and their immunotherapeutic potential in cancer treatment”

Salazar et al.

 In this review, the authors provide a comprehensive summary of the communication between tumor cells and their microenvironment, focusing on the strong immunological responses triggered in tumor tissues after radiation therapy. The authors also discuss the nature of the Abscopal effect, which is a phenomenon where localized radiation therapy to a tumor can lead to a systemic antitumor response at distant sites. The review also highlights recent discoveries in cancer cell communication via extracellular vesicles and their potential use as an immunotherapeutic treatment against cancer.

The article is not only well-written, but also presents the information in a clear and concise manner, with the aid of informative tables and exceptional figures that help to illustrate the authors' ideas. Chapter 5 (Modifying extracellular vesicles for therapeutic purposes) stands out for its scientific value, by revealing the therapeutic potential of extracellular vesicles in cancer therapy.

Overall, this review article is of exceptional quality and provides valuable insights into the topic of cancer cell communication and its implications for cancer therapy. Based on the quality of the review, I highly recommend its publication in the journal.

Author Response

We would like to thank to the reviewer for the kind words to our manuscript.  

Reviewer 2 Report

The review article ‘Abscopal effect, extracellular vesicles and their immunotherapeutic potential in cancer treatment’ was well received and has been reviewed. The article seems well written. The authors in this article has given an introduction about abscopal affect and then established through already existing data about how does abscopal affect leads to release of extracellular vesicles. Then the role of extracellular vesicles in cancer and immune system was described.

Here are some suggestions.

The headline ‘Interaction of tumor-derived extracellular vesicles with immune system’ , the authors are advised to make a separate headlines for the role of EVs in immunosuppressive cells and the role of EVs in T, DCs, NK, and B-cells. or it would be better to make a separate sub-heading for each cell type. 

Apart from this, the paper looks good. The authors can revise as advised and submit the revised version. 

Reviewer 3 Report

The authors attempted to assess the relationship between the abscopal effect and EVs in the manuscript. The topic is exciting and current, and the concept that EVs determine this phenomenon is fascinating.

Authors are requested to consider the following comments:

1. Abscopal effect should be described in more detail.

2. All abbreviations should be explained when used for the first time.

3. The authors described the role of EVs in tumorigenesis. However, the part of these structures in the abscopal effect is neglected. This aspect of the manuscript needs to be described in more detail.

4. How did the authors decide what they would cover and in what order in the subsections of the manuscript? Do these pieces of paper relate to the main topic of the manuscript?

5. References must be extended:

doi: 10.3390/cells11182913.

doi: 10.3390/cancers13040847.

doi: 10.3390/cancers12123563.

doi: 10.3390/ijms22179586.

doi: 10.3390/ijms21155373.

doi: 10.3390/ijms18061122.

doi: 10.3390/cells10123429.

doi: 10.3390/cells11030490.

Minor grammatical errors.

Round 2

Reviewer 3 Report

The authors have satisfactorily addressed most of my concerns.